# Design of nanobody targeting SARS-CoV-2 spike glycoprotein using CDR-grafting assisted by molecular simulation and machine learning

Matheus V.F. Ferraz [1,2,3,¤#a*], W. Camilla S. Adan[1,2,¤#b], Tayná E. Lima[1], Adriele J.C. Santos[4], Sérgio O. de Paula[4], Rafael Dhalia[1], Gabriel L. Wallau[5,6,7], Rebecca C. Wade[3,8,9], Isabelle F.T. Viana[1,6], Roberto D. Lins[1,6]

**a** Department of virology, Aggeu Magalhães Institute, Oswaldo Cruz Foundation, Recife, Brazil, **b** Department of fundamental chemistry, Federal University of Pernambuco, Recife, Brazil, **c** Molecular and Cellular Modeling group, Heidelberg Institute for Theoretical Studies, Heidelberg, Germany, **d** Department of General Biology, Federal University of Viçosa, Viçosa, Brazil, **e** Department of Entomology, Aggeu Magalhães Institute, Oswaldo Cruz Foundation, Recife, Brazil, **f** Fiocruz Genomic Network, Oswaldo Cruz Foundation, Recife, Brazil, **g** Department of Arbovirology, Bernhard Nocht Institute for Tropical Medicine, WHO Collaborating Center for Arbovirus and Hemorrhagic Fever Reference and Research. National Reference Center for Tropical Infectious Diseases, Hamburg, Germany, **h** Center for Molecular Biology (ZMBH), DKFZ-ZMBH Alliance, Heidelberg University, Heidelberg, Germany, **i** Interdisciplinary Center for Scientific Computing (IWR), Heidelberg University, Heidelberg, Germany

¤ These authors contributed equally to this work.
#a Current Address: NEC OncoImmunity AS, Oslo, Norway
#b Current Address: Institute of Chemistry, University of São Paulo, São Paulo, Brazil
* matheeusferraz@gmail.com (MVFF); roberto.neto@fiocruz.br (RDL)

## Abstract

The design of proteins capable effectively binding to specific protein targets is crucial for developing therapies, diagnostics, and vaccine candidates for viral infections. Here, we introduce a complementarity-determining region (CDR) grafting approach for designing nanobodies (Nbs) that target specific epitopes, with the aid of computer simulation and machine learning. As a proof-of-concept, we designed, evaluated, and characterized a high-affinity Nb against the spike protein of SARS-CoV-2, the causative agent of the COVID-19 pandemic. The designed Nb, referred to as Nb Ab.2, was synthesized and displayed high-affinity for both the purified receptor-binding domain protein and to the virus-like particle, demonstrating affinities of 9 nM and 60 nM, respectively, as measured with microscale thermophoresis. Circular dichroism showed the designed protein's structural integrity and its proper folding, whereas molecular dynamics simulations provided insights into the internal dynamics of Nb Ab.2. This study shows that our computational pipeline can be used to efficiently design high-affinity Nbs with diagnostic and prophylactic potential, which can be tailored to tackle different viral targets.

**Data availability statement:** All the input data, scripts, and data sets necessary to recreate the

analysis, machine learning models, and computer simulations described in this work are available at https://github.com/mvfferraz/NbDesign.

**Funding:** M.V.F.F. was supported by the CAPES/DAAD bi-nationally supervised doctoral degree (Personnel–Research grant number: 57507871). This work was supported by the State of Pernambuco Funding Agency (FACEPE) under grant numbers BFP-0010-2.11/22 to IFTV and APQ-0346-2.09/19 to RDL; the National Council for Scientific and Technological Development (CNPq) under grant numbers 165071/2018-4, 303833/2022-0, 425997/2018-9 and INCT-FCx to RDL; the Oswaldo Cruz Foundation through the Innovation Program (INOVA) under grant numbers VPPCB-007-FIO-18-2-134 and IAM-005-FIO-22-2-44 to IFTV and RDL; and Fundação Oswaldo Cruz (grant number: VPGDI-050-FIO-20-2-13-36 (Genomic Surveillance Network) to GW. The funders had no role in study design, data collection and analysis, decision to publish, or preparation of the manuscript.

**Competing interests:** MVFF is currently employed by NEC OncoImmunity AS. Other than that, the authors have declared that no competing interests exist.

## Author summary

In this study, we present a pipeline for designing a high-affinity nanobody (Nb) targeting the SARS-CoV-2 spike protein using enhanced sampling molecular dynamics simulations and CDR-grafting. To address the challenges of CDR grafting in Nbs, including the need for structural similarity between the CDR motif of interest and the scaffold region, we utilized the Nb scaffold cAbBCII-10, known for its versatility in accommodating various CDRs. We generated a library based on the cAbBCII-10 framework with diverse, unrelated CDRs and applied machine learning to identify the most promising candidates. Our approach enabled successful engineering of a Nb that binds to the SARS-CoV-2 spike protein with high affinity, demonstrating the effectiveness of our design pipeline for potential therapeutic applications.

## Introduction

With four approved therapies and several others in clinical trials [1,2], nanobody (Nb) technology, a robust and stable new generation of antibody-derived biologics [3], is set to transform the biopharmaceutical industry. This industry, which includes monoclonal antibodies, hormones, and recombinant vaccines, is projected to see global revenue surge to an estimated US$ 567 billion by 2030 [4], underscoring the strong demand for biopharmaceuticals. Due to their simple structures, stability, and ability to be produced rapidly, Nbs have gained attention for their versatility and strong affinity to targets [3]. These features allow their application in areas such as inflammation, oncology, and infectious diseases [5–7]. However, despite the enormous pharmaceutical interest, discovering novel Nbs remains a challenging task endeavor. It requires extensive laboratory screenings to ensure specific targeting, a process that is inherently time-consuming [8,9]. Moreover, screening procedures commonly prioritize the tightest binding Nbs, typically linked to immunodominant epitopes. This focus can unintentionally overlook antibodies that, despite their lower affinities, bind to functionally important sites [10]. Additionally, there is a risk that viruses may adapt by eliminating these immunodominant epitopes, as observed in the evolution of SARS-CoV-2, potentially rendering the antibodies ineffective [11].

Computational design of Nbs offers a promising solution for overcoming these limitations by significantly reducing the associated time and costs [12]. It allows for a highly controlled screening of desired biophysical properties and enables the precise targeting of specific epitopes of interest. However, designing proteins for specific targets is complex and often results in low success rates, requiring multiple rounds of affinity maturation experiments [10,13]. Protein design challenges often stem from inadequate energy functions used to compute free energies [14,15], which typically overlook factors like entropy and molecular flexibility because they are used on static models representing only a single state [16]. This is particularly relevant for Nb design, as Nbs often have extended and highly flexible loops, known as complementary determining regions (CDRs). Given that proteins exhibit marginal stability, typically with free energies of unfolding in the range of 5-15 kcal/mol, accurately computing these energies is essential for precision protein design. Despite advances in data-driven methods [17,18], including deep learning, which translate massive protein-related data into complex function approximations in high-dimensional spaces, their effectiveness in creating functional proteins across various protein targets remains limited. This limitation is partly due to the influence of diversity and quality of the training data on the generalization of deep

learning-based methods [19], underscoring the importance of more realistic energy functions and improved sampling algorithms.

A common protein design technique involves grafting functional motifs or hot spots, which are crucial for binding, onto different protein scaffolds [20–22]. This method transfers binding specificity by incorporating these critical regions, starting with functional residues that already exhibit favorable interactions, and increasing the likelihood of achieving the desired binding properties [23]. This technique has been widely applied across various proteins, including antibodies (Abs) and Nbs, to humanize Abs and their fragments from different organisms for use in human therapeutics [24,25]. More recently, grafting has been employed to transfer specificity in Abs/Nbs, particularly through the grafting of CDRs [26].

However, a limitation of grafting methods is the requirement for a scaffold that can tolerate residue transfer, which is limited by the availability of suitable scaffolds [23]. This is especially challenging for Nbs because their regions of interest often correspond to loops, which are flexible and less structured compared to other protein regions. These loop regions can be more difficult to graft onto other scaffolds without compromising their functional integrity, as their inherent flexibility may not be supported by all scaffold types. Additionally, in Nbs, grafting CDRs and introducing mutations in the framework often leads to loss of function and stability [27–29]. To address this issue, Vincke and collaborators [28] proposed a universal framework called CabBCII-10, identified as able to accept CDR grafting without compromising stability. Initially intended for the humanization of Nbs, CabBCII-10 presents a promising solution. In this work, we hypothesize that this scaffold could also be used to design Nbs capable of binding any target through CDR/hot spots grafting, followed by computational redesign to enhance binding properties. To overcome the conformational limitation of the CDR loops within this universal framework structure, we propose creating a library of the CabBCII-10 framework grafted with CDR loops from various non-congeneric and unrelated Nbs.

Therefore, to address the grafting of hot spots in Nbs' scaffolds, we propose a comprehensive Nb design pipeline (Fig 1A). This pipeline leverages the initial structures of known interactions in antibody-antigen complexes, grafting these interactions onto the universal framework surface. To better understand the relevant interactions for grafting, we employ enhanced sampling molecular dynamics simulations to capture relevant timescales. As a proof of concept of our approach, we have utilized our protocol to design Nbs that specifically target SARS-CoV-2, the virus responsible for the global COVID-19 pandemic [30]. In particular, we have designed Nbs to specifically target the receptor binding domain (RBD) of the virus, aiming to prevent it from binding to the human receptor angiotensin-converting enzyme-2 (hACE2) and thereby inhibiting the mechanism of cell entry.

Experimental measurements show that this designed Nb binds with high affinity to the RBD and to the virus, thus validating our computational protocol for Nb design.

## Materials and methods

### Molecular dynamics and metadynamics simulations

The crystal structure of the SARS-CoV-2 RBD bound to the Ab CR3022 (PDB ID: 6W41) [31] and of the Nb CabBCII-10 (PDB ID: 3DWT) [28] were refined using the Rosetta FastRelax protocol [32,33] with coordinate constraints applied to the heavy atoms. The protonation states of residues were computed at a pH of 7.4 using the PROPKA code [34]. All simulations were performed using GROMACS 2022.5 [35] utilizing the Amber ff14SB force field

[36] for proteins, GLYCAM06 [37] for carbohydrates, and the TIP3P water model [38]. Glycans were added via CHARMM-GUI web server (https://www.charmm-gui.org/) [39,40] following mass spectroscopy data obtained by Crispin and coworkers [41] (S1 Fig). The systems were centered in cubic boxes with periodic boundary conditions, and Na$^+$ and Na$^-$ ions were added to achieve electroneutrality and a saline concentration of 150 mM. Long-range electrostatics were handled with Particle Mesh Ewald [42], and Lennard-Jones interactions had a 12 Å cutoff. After 30,000 steps of energy minimization, the systems were heated to 303.15 K for 100 ps in the NVT ensemble and equilibrated in the NPT ensemble using Langevin thermostat with a collision frequency of 1.0 ps$^{-1}$ and Parrinello-Rahman barostat with a relaxation time of 2 ps [43,44] for temperature and pressure control, respectively. The LINCS algorithm [45] was used to constrain bonds involving hydrogen atoms. For the production phase, all simulations were carried out for 1 μs using a uniform 4-fs time step achieved through the use of hydrogen mass repartitioning [46].

The last snapshot from the conventional MD simulation was used as the starting coordinates for metadynamics [47] simulations, which were carried out using the same version of Gromacs interfaced with the Plumed 2.5 [48] plug-in. Four collective variables were employed based on Löhr et al [49]: the distance between the instantaneous values of a set of torsional angles for the backbone and side-chain and a reference value of $\pi$ radians (180 degrees), the alpha helical content of the CDRs, and the antiparallel beta sheet content of the CDRs. A bias factor of 24 was used. Gaussians were added every 500 ps with a height of 1.2 kJ/mol. The following Gaussian widths were used for the four aforementioned CVs, respectively: 1.5 nm, 0.5 nm, 2.0,1.5. All these values were extracted from [49]. Three replicates simulations were carried out under NPT ensemble conditions for 100 ns with the same parameters as the conventional MD simulations.

## Hot spots mapping

The ten structures with the lowest Gibbs free energy, identified from the reconstructed free energy landscape using metadynamics simulations, were selected to calculate the residue contributions to the binding interface formation. This calculation was performed using alanine scanning (AS) mutagenesis through the Robetta webserver (http://old.robetta.org) [50,51]. Residues within the interaction interface were considered based on the following criteria: 1) having at least one atom within a 4.0 Å distance from an atom in the opposite interface, or 2) being buried in the interface upon complex formation. Hot spots for grafting onto the surface of the CabBCII-10 Nb were determined based on alanine mutations that resulted in a $\Delta\Delta G$ greater than 1.0 kcal/mol,

## Generation of a Database of CDRs grafted into CabBCII-10 Nb

A synthetic library using the CabBCII-10 framework (PDB ID: 3DWT) containing various CDRs was generated. To create this database, the CDRs in the CabBCII-10 Nb were replaced with all available CDRs from the PyIgClassify (http://dunbrack2.fccc.edu/PyIgClassify/) as of May 1, 2021. For this purpose, RosettaAntibodyDesign (RAbD) [52] was used to graft the CDR loops randomizedly. Following grafting, side-chain and backbone geometry minimization were carried out. Subsequently, different artificial structures containing the CabBCII-10 Nb framework alongside CDRs of various other Nbs were used for hot spots grafting. The identified hot spots were grafted onto different potential CDR loops within the framework. The final library comprises nearly 80,000 synthetic Nbs with random CDRs on the CabBCII-10 framework. No redundant sequences were observed at a 95% sequence identity threshold.

## Protein design

Starting with the generated synthetic library containing the CabBCII-10 Nb with diverse CDRs, the identified hot spots were structurally aligned to all proteins in the library to determine the best carrier and region for transplantation using the MotifGraft mover [53] from the Rosetta package [54]. To accept an alignment, the root mean-square deviation (RMSD) between the backbone of the hot spots and the scaffold protein should be lower or equal to 1 Å. After transplantation, the carrier protein was redesigned within an 8 Å radius around the binding interface to enhance favorable interactions and ensure proper folding of the new protein.

Using the Rosetta package, 500,000 novel sequences for the Nb were generated using Monte Carlo combinatorial mutations. Mutations were only permitted in the CDR loops—excluding the hot spots—and not in the framework regions. Disulfide bonds, particularly those between the CDR loops H1 and H3, are known to contribute to the stability of nanobodies. As these bonds were present in the original nanobody we built upon (PDB ID: 3DWT), they were preserved in the grafted designs to maintain structural integrity and stability. The design process employed the *ProteinInterfaceDesign* subroutine, involving four cycles of design and energy minimization. These cycles alternated between "hard" and "soft" energy functions, the "hard" energy function being the default all-atom Rosetta function with strong repulsion forces, and the "soft" function, known as "soft-repulsive" (soft-rep), focusing less on van der Waals overlaps and the strain from residue conformational changes. This was followed by additional side-chain optimization. Additionally, a position-specific-scoring matrix (PSSM) was created for the parental Nb sequence to prioritize mutations with higher PSSM scores during the design process. Only amino acid identities with PSSM score greater than zero were allowed at each position [55]. For all applications using the Rosetta package, the energy function Ref2015 [56] has been used.

## Filtering of sequences

Nbs with potential to bind the S protein were selected based on Rosetta interface metrics which compared their bound and unbound states. These parameters were informed by a dataset of approximately 80 natural Nb-antigen interfaces described by Zavrtanik and Hadži (2019) [57] (S2 Fig). Complex structures were refined using FastRelax, and interface properties were computed via the InterfaceAnalyzer (S1 Script in S1 Text). The initial selection criteria included: shape complementarity > 0.7, binding and interaction energies < -30 Rosetta Energy Units (REU), binding energy density < -2.0 REU/Å, fewer than 10 unsatisfied hydrogen bonds, and RMSD < 1.0 Å post-FastRelax. Interfaces were visually inspected to exclude any buried charged residues or an excessive number of alanine residues.

As a second filtering step, a machine learning (ML) model was developed to differentiate between high and low/medium affinity complexes using 47 complexes from the Zavrtanik and Hadži dataset with known binding free energies in kcal/mol. A threshold of -11.2 kcal/mol was employed to ensure class balance. The model was trained using linear discriminant analysis followed by k-nearest neighbour methods with uniform weights and Euclidean distance, implemented using Python's scikit-learn [58]. Further details on model training and evaluations are provided in S1 Methods in S1 Text.

## Production of recombinant RBD and SARS-CoV-2 virus-like particles (VLPs)

The plasmid for expressing the SARS-CoV-2 receptor-binding domain (RBD) was kindly provided by Dr. Florian Krammer (Mt. Sinai School of Medicine). Recombinant RBD

expression and purification were carried out as previously described [59]. Briefly, HEK 293 cells were transfected with the Wuhan-Hu-1 spike glycoprotein gene RBD fused to a C-terminal 6x-HisTag. After 3 days, the cells were harvested by centrifugation and the supernatant was loaded onto a HisTrap HP column (Cytiva) equilibrated with buffer A (50 mM Tris–HCl pH 8.0, 300 mM NaCl, 20 mM imidazole). The protein was eluted with buffer A supplemented with 300 mM imidazole, and fractions were evaluated by SDS-PAGE with Coomassie staining.

For the production of virus-like particles (VLPs), Vero E6 cells were transfected with plasmids encoding the SARS-CoV-2 structural proteins (M, E, S, and N) in DMEM medium with 10% fetal bovine serum. Using the FuGENE HD E2311 kit (Madison, Wisconsin, USA) as a transfection reagent, following the manufacturer's instructions. This kit was used for transfection, as it proved to be efficient for the experiment, as observed in the work of Shum and Djaballah [60]. Each plasmid was used with a different molarity, according to the protein it encodes [61]. At 24 or 48h after transfection, the cells were removed by centrifugation at 1000 rpm for 10 min at 4°C. The supernatant was layered onto a 20% (w/v) sucrose cushion and ultracentrifuged for 7 h at 21,000 x g at a temperature of 4°C. Pellets containing VLPs were resuspended in TNE buffer [50mM Tris-HCl, 100mM NaCl, 0.5mM EDTA (pH = 7.4)]. SARS CoV-2 VLPs were placed onto a carbon-coated grid for 1 min, stained with phosphotungstic acid (Sigma) for 45 s and air-dried overnight. Samples were examined using transmission electron microscopy (Zeiss, Libra 120). Purified VLPs were separated on 4-20% SDS-polyacrylamide gels and resolved proteins were subsequently transferred to a nitrocellulose membrane for Western blot analysis. A pool of serum collected from patients infected with SARS-CoV-2 at least 30 days after infection was used as the primary antibody, and horseradish peroxidase (HRP)-conjugated anti-human IgG (Jackson ImmunoResearch) was used as a secondary antibody. The bands were visualized using the chromogenic substrate 3,3'-Diaminobenzidine (DAB).

## Nb Ab.2 expression and purification

The gene encoding the Nb Ab.2 was synthesized and inserted into the pET-22b(+) expression vector at the NdeI and XhoI restriction sites, in frame with an N-terminal pelB signal peptide, by GenScript (USA). At the C-terminal end, Nb Ab.2 was linked to a llama linker, followed by an AviTag, a TEV cleavage site, and a 6xHis-Tag. Although the biotin acceptor sequence was incorporated, the Nb was not biotinylated in this study. The resulting construct was transformed into E. coli One Shot BL21 Star (DE3) (Invitrogen) using a standard heat-shock procedure. Cells were grown in LB medium supplemented with 100 μg/ml ampicillin for 16-18 hours at 37°C with orbital shaking at 225 rpm. Protein expression was induced with 1 mM IPTG when the culture reached an OD600 between 0.5 and 0.8, followed by 16-18 hours of incubation at 22°C with orbital shaking at 225 rpm. Cultures were centrifuged at 7000 xg for 30 minutes at 4°C, then resuspended in buffer A (50 mM Tris-HCl pH 8.0, 300 mM NaCl, 20 mM imidazole, 10% (w/v) glycerol) with protease inhibitors. Cells were lysed by ultrasonication, and the lysate was clarified by centrifugation at 48,400 xg for 30 minutes at 4°C. The supernatant was loaded onto a HisTrap HP (Cytiva) column, and proteins were eluted using buffer A supplemented with 300 mM imidazole. Eluted fractions were analyzed by SDS-PAGE, and protein concentration was measured using a spectrophotometer. The purified protein was concentrated using an Amicon Ultra-4 centrifugal filter unit––3000 NMWL (Merck-Millipore).

## Circular dichroism

Circular dichroism (CD) spectra were recorded in the Far-UV region (190-260 nm) using a J-1100 spectropolarimeter (Jasco) equipped with a Peltier-temperature controlled cell holder. The protein was buffer-exchanged into PBS pH 7.4 and concentrated to a final concentration of 4 µg/mL. Measurements were carried out using a 1 mm path length quartz cuvette at a scanning speed of 20 nm/min, and data were averaged over three accumulations. The collected data were deconvoluted using BeStSel algorithm (https://bestsel.elte.hu/sexamin.php) to predict the secondary structure content of the Nb [62,63]. Data visualization was performed using GraphPad Prism version 7.

## Microscale thermophoresis (MST)

Starting from a concentration of 5.8 µM, a 16-point 1:1 serial dilution series of Nb Ab.2 was prepared in PBS (pH 7.4) containing 0.05% Tween-20. Each dilution was then mixed in equal proportions with labeled RBD to achieve a final concentration of 100 nM. The mixtures were incubated for 30 minutes at room temperature (RT). Standard-treated Monolith capillaries were loaded with the sample, and measurements were conducted in triplicate using the Monolith NT.115 equipment (NanoTemper). The measurements were performed at RT, with an LED excitation power of 20% and MST power of 40%. For affinity measurements using SARS-CoV-2 VLPs, a 16-point 1:1 serial dilution series was prepared in PBS (pH 7.4) with 0.05% Tween-20 and combined with labeled Nb Ab.2 at a final concentration of 100 nM. After a 30-minute incubation at RT, the measurements were again performed in triplicate, utilizing an LED excitation power of 60%. The data were fitted to a dissociation constant ($k_D$) model, assuming a 1:1 stoichiometry, using PALMIST software [64] at the T-jump Zone. Data visualization was conducted using the GUSSI software [65]. The binding between SARS-CoV-2 VLPs and Nb-AB.2 may exhibit heterogeneity in binding stoichiometry. Therefore, using the 1:1 fitting model the binding strength is presented in terms of an apparent affinity dissociation constant ($k_{D_{App}}$).

# Results

## Computational design

To identify residues critical to the binding specificity to the SARS-CoV-2 RBD, relevant interactions from the Ab CR3022 were mapped using a multiscale approach (Fig 1A). Initially, metadynamics simulations were employed to enhance the sampling space of the binding partners. These simulations, using a similar set of collective variables (CVs) to those applied here, have been extensively utilized to capture essential conformational movements within the antibodies CDRs loops [66–69]. It is worth noting that the set of CVs chosen was based on their ability to capture the structural transitions and conformational preferences of the complementarity-determining regions (CDRs) during the metadynamics simulations. Even though CDRs are primarily loop regions, the secondary structure elements (e.g., $\alpha$-helicalor$\beta$-sheet content) were selected as CVs because they provide a coarse-grained representation of the conformational landscape, including potential structural rearrangements that may occur during antigen binding. We have used the same set of CVs as Löhr et al [49], who used metadynamics to assess the dynamics of Nbs' loops. One of the main results Löhr et al [49] show is that structural transitions in CDRs often involve transient $\beta$-sheet formation with the antigen or within the CDR itself, which can influence binding affinity. Capturing this behavior helps elucidate the relationship between structural dynamics and binding mechanisms.

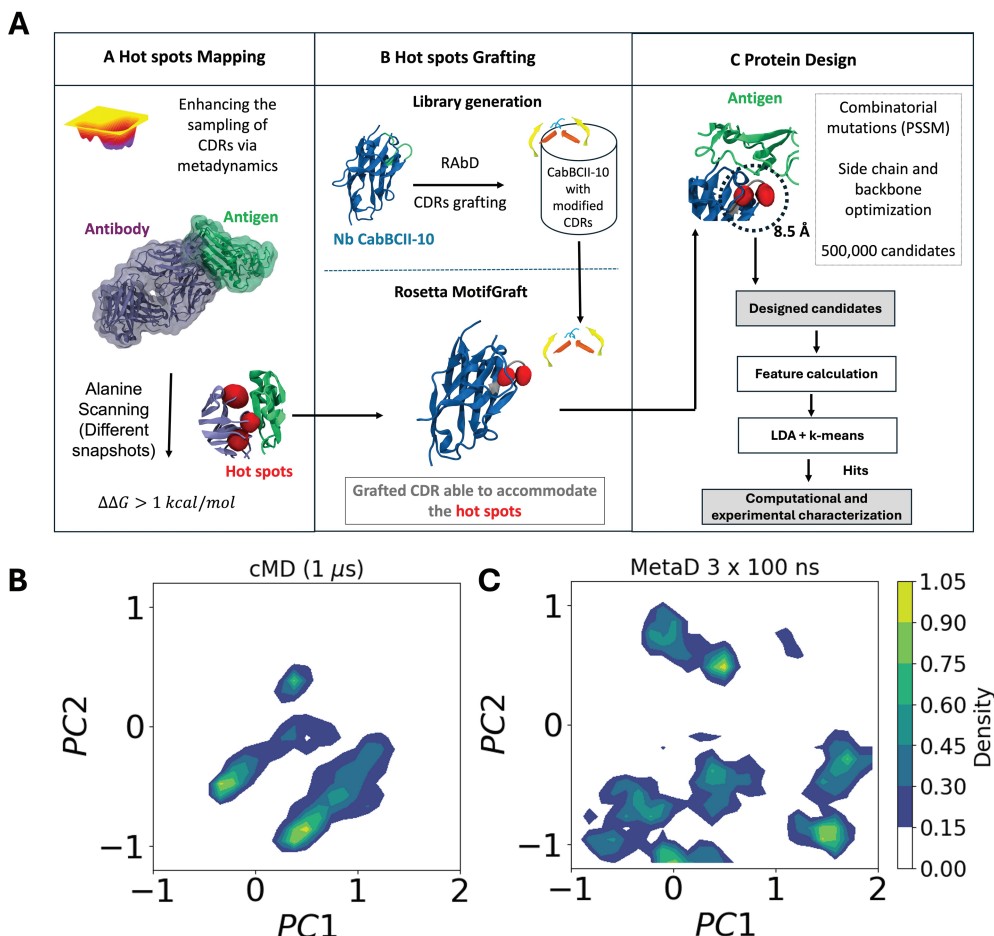

**Fig 1.** (A) Schematic representation of the design pipeline encompassing A. Hot spots (shown as red spheres) mapping by combining enhanced sampling simulations and alanine scanning; B. Hot spots grafting by generating a library containing the Nb CabBCII-10 framework and multiple different CDRs; C. Protein design. (B) and (C) Structural comparison of conformations as a function of the projection on the first two PCA vectors of the trajectory obtained from conventional cMD (1 μs) and metaD (3 replicates of 100 ns), respectively.

Previous studies have demonstrated that enhanced sampling simulations can significantly increase sampling efficiency by several orders of magnitude [70–72]. To evaluate the improvement in sampling achieved by metadynamics relative to conventional MD simulations, we compared the sampled conformational space of the CDR-H3 loop from a 300 ns aggregate metadynamics simulation (comprising from three replicates of 100 ns each) to that from a 1 μs conventional MD simulation. Principal component analysis (PCA), a widely used method for measuring conformational diversity by reducing dimensionality to maximize dataset variance, was employed [73]. In the PCA analysis of the metadynamics and conventional MD simulations, we selected the backbone atoms of the loop H3, which were the areas of interest for our conformational analysis. The analysis was performed on the full trajectory, with no additional re-fitting, except for orientation to ensure consistency in comparison between simulations. The results clearly indicate that metadynamics simulations sampled a substantially larger conformational space, as shown in Fig 1B. This indicates the efficiency of

metadynamics in exploring diverse conformational states that might be missed in conventional MD simulations. We applied the GROMOS clustering method to the conformational ensembles generated by both conventional MD and metadynamics simulations. Using an RMSD cutoff of 2 Å, we identified 3 clusters for conventional MD and 7 clusters for metadynamics, demonstrating that metadynamics samples a broader range of conformational states. Additionally, we calculated the explained variance for the first principal component (PC1) and found that metadynamics captured 68% of the variance, compared to 57% for conventional MD. This indicates that metadynamics not only enhances conformational diversity but also better captures the dominant motions within the system.

Regarding the convergence of the metadynamics simulation, these were relatively short in this study (100 ns per replicate, run in triplicate) to assure converged sampling. However, they were chosen for their ability to rapidly enhance sampling of conformational space, which was critical for our specific objectives. The goal of these simulations was not to achieve fully converged free energy landscapes but to efficiently sample diverse conformational states of the antibody CDR regions to identify key interactions and transitions relevant to binding specificity. We have observed that for this goal, previous studies have successfully utilized short metadynamics simulations to explore conformational transitions in antibody CDRs, even without full convergence, as referenced in our manuscript [97,98]. These studies demonstrated that even short metadynamics runs could reveal biologically relevant structural features and transitions.

Ten structures from the most densely populated regions from the PCA space were extracted and alanine scanning mutagenesis was used by determining alanine mutations that

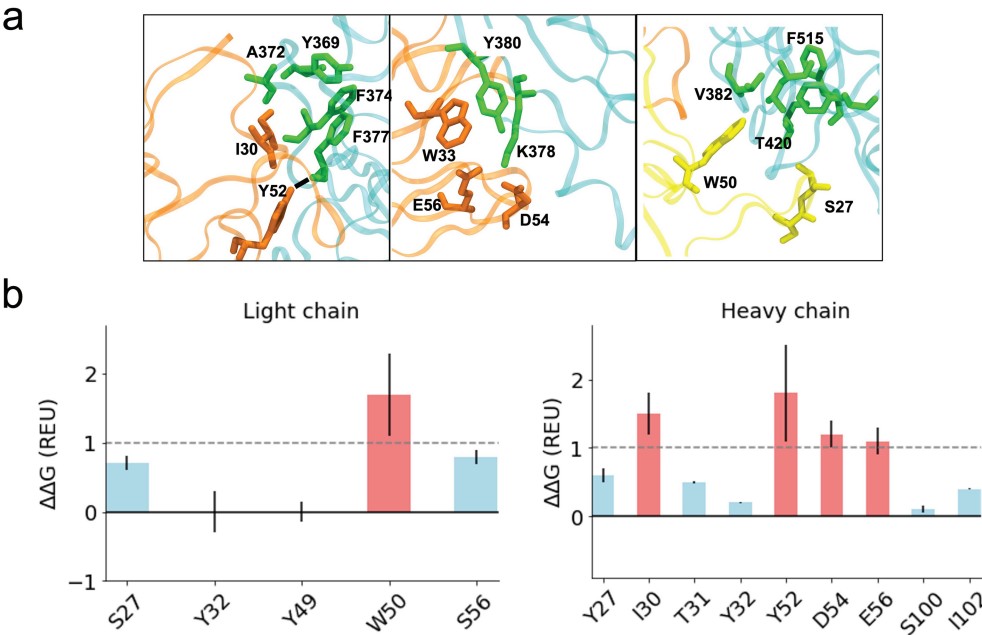

**Fig 2.** (A) Important interactions between CR3022 and the receptor-binding domain (RBD) of SARS-CoV-2 [31]. The heavy chain of CR3022 is depicted in orange, the light chain in yellow, and the SARS-CoV-2 RBD in cyan. Dashed lines indicate the presence of hydrogen bonds(B) Computational mutagenesis by alanine scanning for the residues in the interaction interface. Each bar denotes the predicted binding ΔΔG for a given residue upon alanine mutation. A threshold of 1 REU, shown as a dashed line, was chosen as cut-off to predict destabilizing effects (red bars), suggesting importance for binding. Blue bars represent stabilizing or no effect on the ΔΔG upon alanine replacement.

resulted in a ΔΔG exceeding 1.0 kcal/mol, indicating destabilization of the complex (Fig 2A). The energetic contribution of each sidechain in the protein-protein interface was estimated by the binding ΔΔG. Important residues for the interaction in the Ab were grafted onto the scaffold of an artificial Nb. As it can be seen, the antibody residues that most significantly contribute to the binding to SARS-CoV-2 RBD are located within the heavy-chain domain of the Ab (Fig 2B). While only heavy-chain residues were grafted in this study due to their higher contribution to the binding affinity, incorporating light-chain residues is also viable to enhance binding stability, depending on the specific antigen and desired interactions.

The alanine scanning (AS) result reveals a major role of the heavy chain in binding, highlighting the importance of residues I30 and Y52. I30 is involved in a hydrophobic core, alongside W33, which, although not predicted to be a hot spot, plays a crucial structural role through aromatic interaction, as observed in the crystallographic structure. According to AS calculations, Y52 makes the greatest contribution to the stabilization of the complex. Y52 residue interacts with F377 from the SARS-CoV-2 RBD by forming a hydrogen bond between the hydroxyl hydrogen side-chain of Y52 (donor) and the main-chain oxygen of F377 (acceptor). Moreover, residues D54 and E56 contribute less to the binding energy of the interface, despite being involved in a salt bridge with K378 as observed in the crystallographic structure. Regarding the light chain of CR3022, W50 contributes significantly to the stability of the complex by being part of a hydrophobic core, and S27 makes a hydrogen bond with T430 from the RBD.

Therefore, these residues were also considered for grafting. Since simultaneous grafting of the selected residues from both the heavy and light chains was not possible, we chose to graft the residues from the heavy chain due to their higher contribution. Thus, the following residues from the heavy chain of the Ab were considered important for grafting onto a VHH binding loop: I30, W33, Y52, D54, and E56. These residues were scanned against a library containing different CDRs. The screening resulted in one CDR capable of accommodating the five residues in their native conformation with a backbone atom RMSD of 1 Å. The selected loop corresponds to the H1 loop of the PDB code 5M13 [74], a synthetic VHH complexed with the maltose-binding periplasmic protein. The H1 loop was grafted onto the H1 region of the CabBCII-10 Nb while preserving the scaffold's H2 and H3 loops.

## Sequence filtering

After applying the interface-based metrics filter, only two sequences out of the 500,000 designs were selected (S1 Table). To evaluate redundancy in the generated sequence library, we clustered sequences using CD-HIT at a 95% sequence identity threshold, identifying approximately 120,000 redundant sequences. Although smaller libraries may suffice for certain applications, generating a larger sequence library allows for thorough exploration of the sequence space. Future applications of this method could consider optimizing the library size to balance computational efficiency with design thoroughness. For the second filtering step, we developed a machine learning model to classify complexes into high affinity (< -11.2 kcal/mol) and low/moderate affinity (> -11.2 kcal/mol) categories. We computed 21 Rosetta-interface properties and used an extra tree method to select the 10 most relevant features for the model (S3 Fig). These features along with their definitions are listed in Table 1.

Due to the complexity of distinguishing high-affinity from low/moderate-affinity complexes using individual descriptors, we hypothesized that a comprehensive consideration of all features would enhance classification accuracy, given the multi-dimensional nature of the problem. To test our hypothesis, we applied Linear Discriminant Analysis (LDA) to

**Table 1. Selected Rosetta-calculated interface properties.** Short description of the features based on the Rosetta package energy function. Energies are measured in REU and the features are ordered as a function of the relative importance

| Feature | Description |
|---------|-------------|
| complex_normalized | Average energy of a residue in the entire complex |
| side2_score | Energy of the other side of the interface, here the nanobody |
| hbond_E_fraction | Amount of interface energy (dG_separated) accounted for by cross interface H-bonds |
| dG_separated/dSASAx100 | Separated binding energy per unit interface area * 100 |
| delta_unsatHbonds | The number of buried, unsatisfied hydrogen bonds at the interface |
| side1_normalized | Average per-residue energy on one side of the interface. |
| dG_cross/dSASAx100 | Binding energy of the interface calculated with cross-interface energy terms * 100 |
| dSASA_hphobic | the Hydrophobic interface surface area |
| ddg | Binding energy |
| side1_score | Energy of the first side of the interface, here the antigen |

Definitions were extracted from Rosetta documentation: https://docs.rosettacommons.org/docs/latest/application_documentation/analysis/interface-analyzer

**Table 2. Evaluation of the LDA-KNN classifier using 5-fold cross validation.** Values correspond to the average and standard deviation of the 5 cross validations

| Metric | Value ± SD (5-fold CV) |
|--------|------------------------|
| Accuracy | 0.79 ± 0.11 |
| Precision | 0.82 ± 0.10 |
| Recall | 0.79 ± 0.10 |
| ROC AUC | 0.86 ± 0.07 |
| MCC | 0.86 ± 0.07 |

transform the data and generate scores for distinguishing between high and low/moderate-affinity classes (Fig 3A). The results confirmed our hypothesis: high-affinity complexes clustered together with positive scores, while low/moderate-affinity complexes were predominantly concentrated in the negative values. This observation aligns with the understanding that protein-protein binding is influenced by multiple factors, and that a specific combination of properties is essential for a strong interaction. We analyzed the contributions of each feature to the LDA to identify the energetic factors most influencing Nb-antigen binding (Fig 3B). The total energy of the complex emerged as the most significant factor, with properties such as hydrogen bond energy, binding free energy per solvent accessible surface area, and the normalized energy of the antigen playing key roles.

Despite the clear clustering of high and low/moderate-affinity complexes in distinct regions of the LDA map, some overlap existed at the boundary. To address this, we explored the use of supervised learning to classify the affinity of Nb-antigen complexes accurately. We trained K-Nearest Neighbors (KNN) classifiers, with k=6 (S4 Fig), using 5-fold cross-validation on the selected features. The classifier demonstrated strong performance, as evidenced by the computed metrics (Table 2). Although precision values indicated some misclassification of low/moderate-affinity complexes, the recall metric of 0.79 suggested that only a small proportion of high-affinity complexes were incorrectly classified as low/moderate. The overall accuracy was 0.79, with both the Matthew's correlation coefficient (MCC) at 0.86. These metrics collectively affirm the robustness of our approach. Additionally, the high values in accuracy, MCC, and area under the receiver operating characteristic curve (ROC AUC) (0.86) (Fig 3C) validate the general effectiveness of the K-Nearest Neighbors classifier for this application.

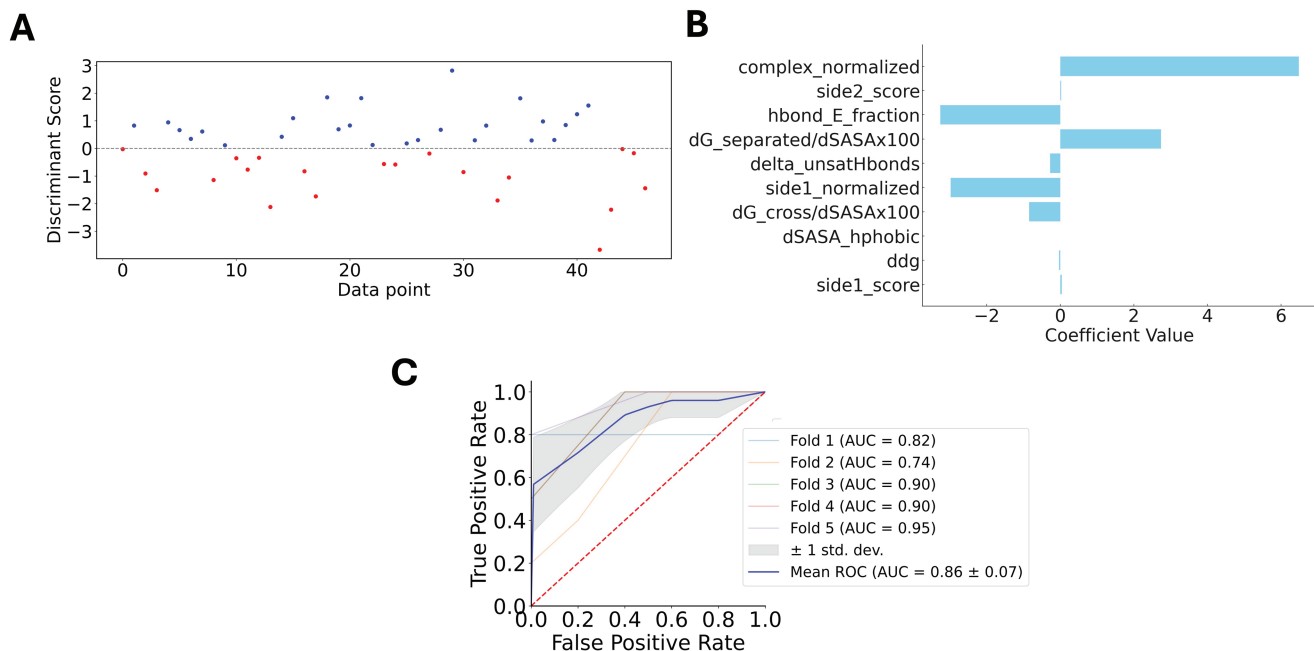

**Fig 3. Machine learning models for the binary classification between high and low/moderate Nb-antigen complex affinities.** (A) Linear discriminant value calculated for each sample in the dataset, demonstrating that the explained variance considering only one LD is 100% (B) Contribution of each feature used to calculate the LDs values (C) Assessment of the classification models' performance through their characteristic ROC curves. Five different curves are shown, each representing one of the 5-fold cross-validations (CVs). The shaded gray area represents the standard deviation across the 5-folds, while the blue line denotes the mean ROC AUC.

## Computational structural and dynamics evaluation

Using the ML-based classification, one of the two filtered sequences was selected as a candidate for experimental assessment, referred to as Nb Ab.2. To evaluate whether Nb Ab.2 adopts the intended 3D structure, we used an unbiased approach with the AlphaFold3 (AF3) server to predict its conformation (Fig 4A). The prediction resulted in an RMSD of 1.5 Å (with a 0.89 Å, 0.45 Å, and 1.38 Å for the CDRs 1, 2, and 3, respectively) and a predicted local distance difference test (pLDDT) score of 90% (with an average of 72%, 91%, and 63%, for the CDRs 1, 2, and 3, respectively), indicating a close match to our design. The lower pLDDT for the CDR3 is not unexpected, as this represents a non-canonical set of loops, and will present a lower sequence coverage in the multiple sequence alignment for this region. Moreover, the loop region in CDR3 is highly flexible. The agreement between the modeled and designed structure is expected, as nanobodies typically adopt a similar fold. Therefore, the AF3 prediction suggests that Nb Ab.2 is likely to fold according to the design.

To investigate structural stability, we conducted MD simulations and compared Nb Ab.2 to the wild-type nanobody, CabBCII-10. Throughout the simulation, Nb Ab.2 exhibited more restricted dynamics compared to its wild-type counterpart, indicating that the designed nanobody is structurally stable and does not undergo major structural changes. Interestingly, while the overall flexibility profiles, measured by the root mean square fluctuation (RMSF), were similar, CDR1 and CDR3 regions in Nb Ab.2 displayed increased flexibility (Fig 4B). This increased flexibility could be attributed to the differences in chemical composition, as

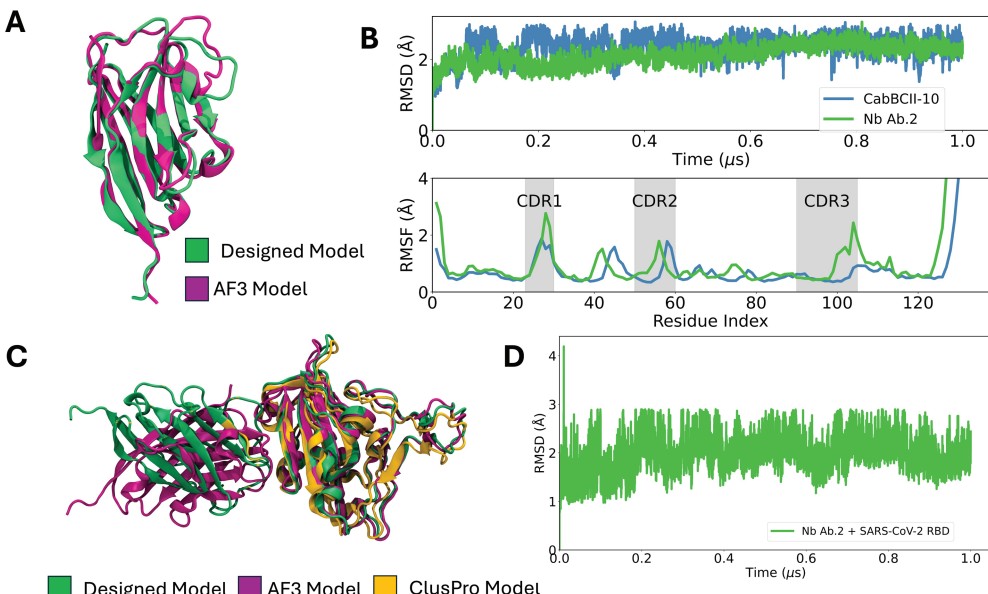

**Fig 4. (A) Structural alignment between the designed structure (green) and the AF3 model (purple).** (B) Time-series properties obtained from MD simulations. top: Root mean square deviation (RMSD) as a function of time between the alpha carbons from the simulated structural ensemble and the crystallographic structure (CaBCII-10, PDB ID: 3DWT) and the modeled structure (Ab.2). Bottom: per-residue root mean square fluctuation (RMSF) for the alpha carbons calculated for the last 0.5 μs of simulation. The shaded gray area represents the CDRs 1-3 regions. The blue line corresponds to CaBCII-10, while the green line corresponds to Nb Ab.2. (C) Structural alignment between the designed complex model (green), AF3 (purple) and ClusPro (orange) predicted models. (D) RMSD for the designed complex Nb Ab.2 and SARS-CoV-2 RBD over time, represented by the green line. All the structural models are depicted in cartoon representation.

CabBCII-10 originally binds to $\beta$-lactamase, and the repurposing for a different binding target altered the specificity and dynamics of this loop. In addition, it has been shown that rigidification of CDRs can also decrease antibody affinity, and considering the role of conformational entropy, these factors must be considered and optimized during the development process of therapeutic antibodies [49].

To investigate whether the designed Nb binds to the target region in the SARS-CoV-2 RBD, we performed molecular docking calculations using the antibody mode of the ClusPro web server [75]. ClusPro has demonstrated reliable results for protein-protein global docking, as evidenced by its successful performance in the critical assessment of predicted interactions (CAPRI) experiments [76]. The ClusPro docking generated multiple complexes with the Nb positioned at various sites on the SARS-CoV-2 RBD, including the intended target region. The docking pose with the lowest energy had an RMSD of 0.8 Å compared to the structure of the Rosetta-designed complex (Fig 4C).

Unlike other docking methods that rank structures solely based on energy, ClusPro typically ranks them by cluster size, which often corresponds to the lowest energy cluster [77]. In our docking calculations, however, the lowest energy cluster was the seventh largest. To determine whether the largest cluster or the lowest energy cluster represents the most stable conformation, we calculated the binding free energies using the molecular mechanics generalized Born surface area (MM-GBSA) method [78] for both models (S2 Methods in S1

Text). The MM-GBSA calculations indicated that the seventh-largest cluster had a more stable conformation ($-34.02 \pm 7.79$ vs. $11.88 \pm 3.48$ kcal/mol; see S2 Table for energy components breakdown). This suggests that the designed complex binds to the SARS-CoV-2 RBD as intended.

For further in silico validation of the designed complex structure, we used AF3 to predict the structure of the complex. Using the default setup, the predicted AF3 structure showed good agreement with the Rosetta output, presenting a 2 Å RMSD and a similar binding mode at the target site. However, the interface confidence score was 0.55, indicating low confidence in the interface prediction.

MD simulations of the complex formed by Nb Ab.2 and the SARS-CoV-2 RBD were carried out to investigate its stability and binding dynamics. The RMSD values were monitored throughout the simulation to evaluate the conformational stability of the complex. The RMSD results indicated that Nb Ab.2 remained consistently bound to the RBD binding site, with minimal deviations, suggesting a stable interaction (Fig 4D).

## Experimental characterization

Nb Ab.2 was bacterially expressed and purified using a chromatography system (S5 Fig A). The purity was confirmed by acrylamide gel electrophoresis (SDS-PAGE) (S5 Fig B). This designed Nb is formed by 176 amino acids and has a theoretical molecular mass of 19.3 kDa, and an expression yield of 0.89 mg/mL. ELISA was carried out for detection (S3 Methods in S1 Text; S5 Fig C).

Circular dichroism (CD) spectroscopy was performed to evaluate the secondary structure profile of the purified Nb Ab.2. The CD spectrum revealed a negative minimum at 218 nm corresponding to the antiparallel beta-sheet, which is indicative of the core immunoglobulin fold [79] (Fig 5A). These findings are consistent with previously reported CD spectra of similar nanobody structures, which typically exhibit such features due to the $\pi-> \pi*$ transitions in the peptide bonds. The secondary structure profile observed experimentally qualitatively corroborates the predicted secondary structure content from 1 μs molecular dynamics simulations, which also indicate a high $\beta$-sheet content (Fig 5B). This agreement between experimental and computational data validates the structural integrity of Nb Ab.2, confirming that it folds correctly.

The quantitative analysis of secondary structure content was performed using the BeStSel method [63], which provides precise quantification of beta-sheet content, distinguishing between parallel and anti-parallel beta-sheets. This is essential for Nbs, whose stability relies heavily on well-defined beta-sheet structures. The Nb is composed mostly of antiparallel $\beta$-sheets (47.3%). Other structural elements account for 33% of the Nb's composition and include the CDR domains. Additionally, BeStSel also classified the Nb as a fragment of immunoglobulin G, formed by sandwich-like $\beta$-sheets. The secondary structure content of Nb-BCII10 was determined using the PDB entry 3WDT as input in the BeStSel software. Accordingly, 3WDT is composed of 40.2% of antiparallel $\beta$-sheets and 47.4% of other elements. The proportion of parallel $\beta$-sheets calculated for Nb Ab.2 (7.3%) was slightly higher than that predicted for the reference structure (4.3%) and is likely attributable to the CD signal observed at 230 nm [80].

Microscale thermophoresis (MST) was utilized to measure the binding affinities in solution. Experimental MST data fitting for the interaction between the RBD and the designed Nb Ab.2 yielded a dissociation constant ($k_D$) of 9 nM (63.8% confidence interval: 6 to 13 nM) (Fig 5C). For SARS-CoV-2 virus-like particles (VLPs, S6 Fig), the measured binding affinity of Nb Ab.2 was 60 nM (63.8% confidence interval: 40 to 80 nM) (Fig 5D).

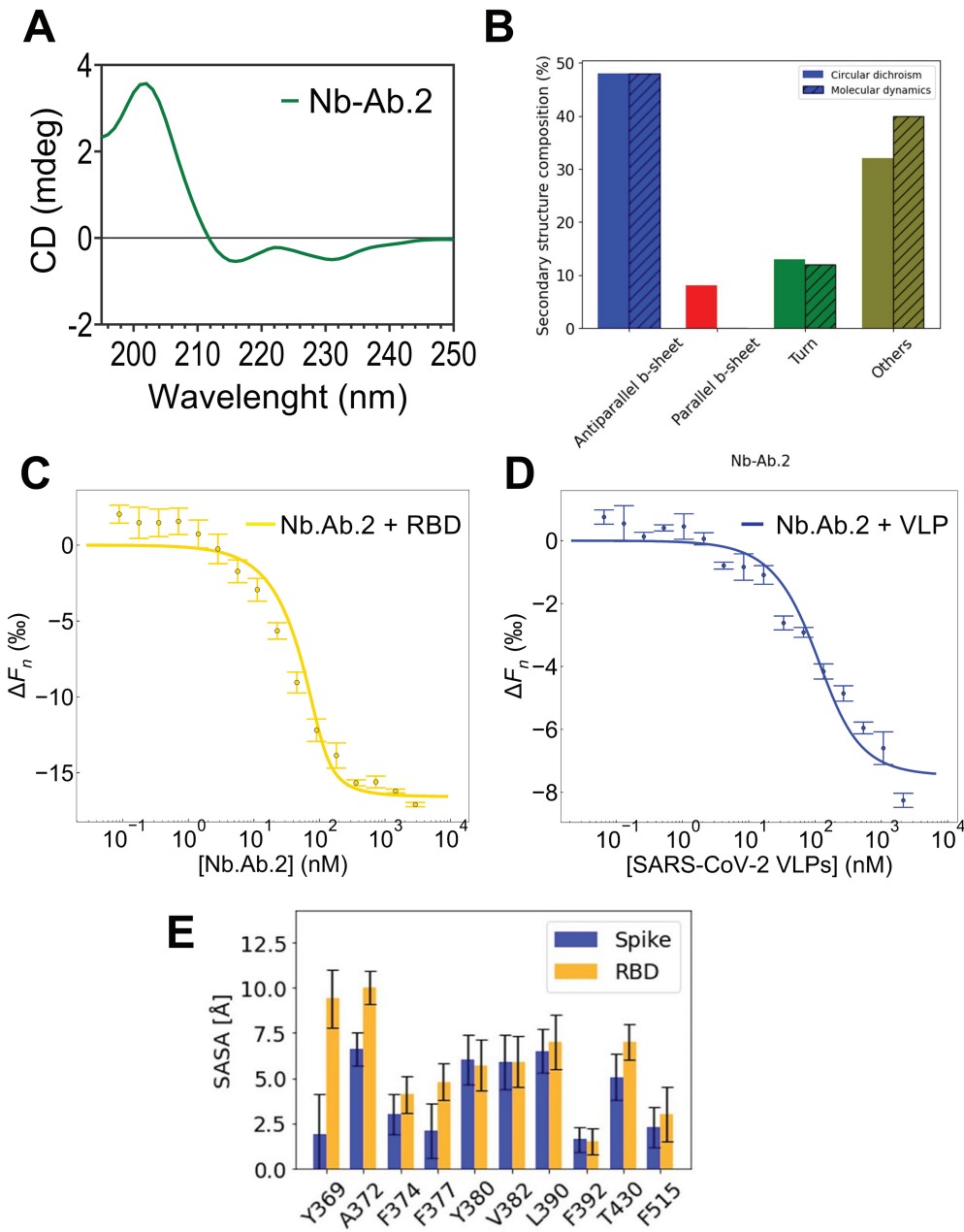

**Fig 5. Biophysical characterization of Nb Ab.2.** (A) Circular Dichroism (CD) spectra of Nb Ab.2 displaying characteristic signals of $\beta$-structured immunoglobulin (IgG) folded domains. (B) Estimation of secondary structure components derived from the CD data using BeStSel. (C, D) left: MST binding curve illustrating the interaction between recombinant SARS-CoV-2 RBD and Nb Ab.2. right: MST binding curve demonstrating the interaction between SARS-CoV-2 VLPs and Nb Ab.2. (E) Solvent-accessible surface area obtained from different snapshots of 200-ns molecular dynamics simulations of the RBD binding site considering the free RBD (orange) and within the fully glycosylated spike protein (blue). The snapshots were recorded at every 1 ns.

When determining $k_D$ according to the law of mass action, it is recommended to use a concentration of the labelled molecule lower than the $k_D$ value. Due to sensitivity limitations, concentrations below the measured $k_D$ were not feasible. However, Monte Carlo simulations

of typical MST experiments, as demonstrated by [12], suggest that reliable determination of $k_D$ values can be achieved even when the $k_D$ value is approximately one-tenth the concentration of the labelled molecule.

The experimentally derived $k_D$ value of Nb Ab.2 binding to SARS-CoV-2 RBD corresponds to a binding free energy of –10.7 kcal/mol at room temperature (293.13 K). To compare this experimental value with structure-based computational models, we utilized an artificial neural network (ANN) model to predict the binding free energy for protein-protein complexes on the basis of the designed structure. This ANN model, which has shown an average error of 1.6 kcal/mol for Nb-antigen complexes [81], predicted a binding free energy of –11.8 kcal/mol. The predicted value is within the method's margin of error, demonstrating consistency between the experimental data and the computational model based on the designed structure.

To ascertain that this affinity is a result of the design protocol, we compared the measured binding free energies using the Rosetta suite energy function. We have superimposed the structure of the WT Nb, which we built our design on, on the structure of the Nb Ab.2 and it displayed a positive binding energy of 13.82 REU, compared to the binding energy of -37.72 for the Nb Ab.2–RBD complex. It is worth noting that for the modeling of the complexes, the same level of theory for relaxing the complexes' structure was employed, suggesting the WT Nb is not expected to bind to the SARS-CoV-2 RBD.

As demonstrated by the MST assays, Nb Ab.2 exhibited high affinity for the isolated SARS-CoV-2 RBD. However, this affinity decreased when compared to the S protein, as represented by VLPs. A plausible explanation for this difference is that the epitope targeted by Nb Ab.2, which also binds the CR3022 antibody, is buried within the S protein. This epitope becomes exposed and available for binding only when two RBDs are in the "up" position, and the targeted RBD undergoes a rotation of approximately 30 degrees [31]. Currently, it is believed that the lack of neutralization of the CR3022 antibody against SARS-CoV-2 is due to its lower affinity to the SARS-CoV-2 RBD compared to the SARS-CoV-1 RBD. However, our assays demonstrate that despite the high affinity of the designed VHH for the isolated RBD, it does not bind as tightly to the S protein within the VLP. This is presumably because of the conformational arrangement of the epitope within the S protein. To test our hypothesis, we carried out 200 ns of MD simulations for both the isolated RBD and the entire SARS-CoV-2 S protein with 2 RBDs in the "up" position (initial coordinates obtained from CHARMM-GUI archive, under the code 6X2B), which is much less common than 1 RBD in the "up" position. Both systems were fully glycosylated. Fig 5E shows that for the specific residues Y369, A372, and F377, a higher solvent exposure is observed in the isolated RBD, suggesting that these residues are not readily accessible for binding in the context of the full S protein. Notably, F377 forms a hydrogen bond with Y52, one of the grafted residues from CR3022. In addition, the residues Y369, A372, and F377 are part of a hydrophobic core that is exposed to solvent in the isolated RBD, indicating they may play a crucial structural role. To improve the binding affinity to the full spike protein, future design iterations could consider using the full spike structure as the input, allowing for a more comprehensive optimization of the Nb's interactions to the specific conformation of the spike protein.

Despite the promising *in silico* validations, it is crucial to emphasize the need for experimental structure determinations to fully validate the structural predictions from our design.

## Discussion

The pipeline proposed in this work builds upon the MotifGraft routine from the Rosetta package. The MotifGraft protocol superimposes residues of interest onto a scaffold protein surface

from a library of potential scaffolds. However, applying this approach to nanobodies (Nbs) presents two main challenges: the fixed structure of the host scaffold and limitations in scaffold selection. CDR (complementarity-determining region) grafting is a common protein engineering technique used to transfer binding specificities to different antibody frameworks, often for stabilizing or humanizing antibodies for medical use. Its application to Nbs has been limited, typically only grafting CDRs from animals within the same taxonomic family. Minor sequence changes in the framework can unpredictably affect stability, sometimes leading to a complete loss of stability. For instance, transferring CDRs to a framework with only four differing residues was enough to prevent the expression of an engineered Nb [82].

To address these issues and capture a broad conformational space, we used metadynamics as an enhanced sampling technique. This approach helps overcome high-energy barriers between distinct conformational states. Previous studies have emphasized the significant role of the CDR loops in driving antigen recognition [83]. Additionally, these CDR loops are known for their high flexibility, which is crucial for effective antigen binding. CDR grafting is a common protein engineering technique used to transfer binding specificities to different antibody frameworks. This method is often employed to stabilize or humanize antibodies for medical use.

To overcome the challenges presented by traditional scaffold selection and fixed structures of host scaffolds, the universal framework scaffold, Nb CabBCII-10, capable of accommodating CDR grafts, was proposed by Vincke et al. ([28]). Instead of using this universal scaffold for humanization purposes, we used it as a starting point for our protein engineering efforts. Our approach includes increasing the number of possible CDR loops for grafting while maintaining the stability of the designed nanobody, an aspect not yet explored through computational means. To bypass scaffold conformation restrictions and expand the possible conformational space of CDR loops for the grafting protocol, we computationally built a synthetic library of Nbs based on Nb CabBCII-10. This was achieved by swapping its original CDRs with all possible CDRs within PyIgClassify using the RosettaAntibodyDesign protocol.

For our pipeline development, we utilized machine learning to differentiate between high and low/medium affinity Nb-antigen complexes. Our findings establish key factors influencing Nb-antigen affinities, identifying crucial descriptors such as hydrogen bond energy, binding free energy per solvent-accessible surface area, and the normalized energy of the antigen. We demonstrated that hydrophobic interaction area and hydrogen bond contribution are critical factors in modulating affinities. Previous research has highlighted the role of hydrogen bonds in Nb-antigen interfaces [84,85]. Additionally, our work shows that the structural configuration and conformational stability of the antigen plays a crucial role in determining affinity. This finding opens new avenues for investigating the impact of antigen mutations on viral immune evasion of Nbs.

Despite the inherent difficulties in generating correctly folded nanobodies, our design led to a structurally consistent Nb fold, as confirmed by CD spectroscopy and MD simulations. CD spectroscopy was employed to analyze the secondary structure content of the nanobodies, ensuring they folded correctly into their expected conformations. Additionally, MD simulations offered detailed insights into the stability and dynamics of the Nb structures over time, further validating the robustness of our designs. These combined methods confirmed that the nanobodies maintained their structural integrity, which is crucial for their function and binding capabilities.

Identifying Nbs with high affinity has been particularly challenging for RNA viruses such as SARS-CoV-2, which continually evolve to evade immunity. Consequently, there is an increasing demand for platforms that can tailor specific Nbs to combat emerging variants

quickly. In this context, computational protein design (CPD) offers an efficient and cost-effective solution for generating proteins with desired functions. During the COVID-19 pandemic, CPD has significantly contributed to developing alternative therapeutics for SARS-CoV-2. While CPD has been used to develop Nbs targeting the SARS-CoV-2 RBD, the *de novo* design of Nbs for specific epitopes remains in its early stages. Currently, most computational engineering efforts for Nbs have focused on mutating known Nb-antigen complexes or performing computational affinity maturation [86–89]. While some studies have proposed designing entire Nb structures de novo, these approaches have yet to undergo experimental validation [90,91], limiting their practical application. Recently, a protocol using deep learning diffusion methods for atomically accurate de novo Nb design has shown promise [92], opening new avenues for designing next generation Nbs with binding poses nearly identical to the crystal structure. However, their designs presented affinities ranging from 78 nM to 5.5 μM. Additionally, deep learning (DL)-based approaches are mostly constrained by the quality of the training set, and despite advancements, DL-generated structures still exhibit a high failure rate [93]. Moreover, only a few computer-designed Nbs have been successfully designed computationally and tested experimentally [94,95], achieving affinities ranging from 120 nM to 7 μM. Our protocol, however, demonstrates robustness in creating high-affinity binders, with affinities comparable to the average affinity of a Nb for its antigen, which is approximately 6 nM. This is comparable to the affinity of monomeric antigen-binding sites (Fab or ScFv) of conventional antibodies for their antigens [96].

Achieving high affinities through computational protein design typically requires multiple rounds of affinity maturation and extensive experimental adjustments. Remarkably, our current pipeline successfully designed an Nb with natural antibody-like affinities using solely computational methods, bypassing traditional experimental affinity maturation techniques. Our computational approach, which includes predictions from ML models and an artificial neural network (ANN), has shown consistency with experimental data. The designed Nb was predicted to have high affinity by these models and was later confirmed by experimental validation, presenting an affinity of 9 nM to the designed target. However, a lower affinity was observed for the whole protein, where the dynamics of crucial residues in the binding site were correlated with this reduction in affinity. Despite this, it still maintains a high affinity for the target, being two fold higher than the original Ab CR3022 (by roughly comparing two different measurements using different methods and conditions).

One limitation of the pipeline introduced in this work is that it relies on the availability of known antibody-antigen interface structures, as it uses the MotifGraft approach. While it is possible to model the interface using tools like deep learning-based methods [101,102], the accuracy of the resulting designs may be lower compared to those based on experimentally determined structures. In the absence of precise structural information, there may be a bias toward immuno-dominant epitopes. In addition, high-affinity binding often requires the integration of multiple hotspot residues across different CDR loops. While the synthetic nanobody library inherently provides combinatorial diversity by randomly grafting CDRs onto the CabBCII-10 framework, there may still be instances where not all hotspots can be accommodated simultaneously. In such cases, the protein design process may require reducing or prioritizing a subset of hotspots. This limitation is inherent to the MotifGraft approach and should be considered when applying this methodology to highly complex binding interfaces.

Following the successful synthesis and characterization of a high-affinity Nb, future efforts will focus on assessing its therapeutic potential and understanding its molecular interactions. The next steps include evaluating the Nb's neutralizing capacity against live virus *in vitro* and

potentially extending these assessments to *in vivo* models to establish efficacy in more complex biological systems. Structural studies, such as X-ray crystallography or cryo-electron microscopy, will elucidate the Nb's binding sites and conformation upon interaction. These studies will provide detailed insights into the molecular mechanisms of Nb interaction and facilitate the process of affinity maturation to enhance binding efficiency and specificity.

The methodologies and insights from this study could significantly influence future biotherapeutics, particularly in designing treatments for various viral pathogens. It is worth noting that many deep-learning methods are available, such as ProteinMPNN [100] and these can accelerate Nb's design. However, it has also been shown that these methods still struggle to effectively capture the unique sequence-structure relationships of Abs/Nbs [99]. In particular, the model showed biases towards overrepresented residues and has difficulty designing variable regions, such as CDR3, which are critical for the functionality of Nb. The computational design and biophysical characterization techniques developed here could be applied to other viral targets, expanding the potential applications of Nbs in medicine. Moreover, the adaptable CDR-grafting strategy used in this project suggests a versatile platform for generating biotherapeutics tailored to combat a wide array of viral infections, enriching both prophylactic and diagnostic toolkits.

## Conclusion

In this work we introduce and apply a design pipeline for nanobodies (Nbs) targeting specific antigens that combines seeded interface design with computer simulations and machine learning techniques. As a proof-of-concept, we designed a high affinity Nb-RBD binding, demonstrating a strong agreement between theoretical predictions and experimental validation. It is worth noting that this designed Nb presented the highest affinity of a *de novo* Nb-antigen binding interface design reported so far. The results reported here should support development of improved Nb biotherapeutics using the MotifGraft approach, effectively bypassing the need for extensive experimental affinity maturation. In addition to contributing to the scientific understanding of Nb engineering, this work sets a precedent for future biotherapeutic development projects aiming to mitigate various infectious diseases. This approach can accelerate the creation of targeted antiviral therapies and other biotherapeutics, potentially addressing a wide array of medical challenges.

## Supporting information

**S1 Text Interface Analyzer XML script and Supplemental methodology for: Machine learning; Molecular mechanics generalized Born surface area (MMGBSA); and ELISA for the nanobody detection**.
(PDF)

**S1 Fig Illustration of the structure of the SARS-CoV-2 RBD-bound N-glycan attached to N- 343.** The N-glycan is composed of three components: N-Acetylgucosamine (depicted by a blue square), mannose (depicted by a green circle), and fucose (depicted by a red triangle). The $\alpha$ and $\beta$ linkage types are represented by A and B, respectively. The numbering between the glycosidic linkages indicates the carbon involved in the bond, where the first number is the carbon number of the first monosaccharide and the second number is the carbon number of the second monosaccharide.
(TIF)

**S2 Fig Box plot for visually displaying the distribution and skewness of the interface parameters in 80 natural Nbs-antigen interfaces showing the data quartiles and averages.** The interquartile range is shown as a solid blue box, where the top and bottom of the box

denote the upper and lower quantile, respectively. The median is depicted as an orange line. Outliers are represented by circles.
(TIF)

**S3 Fig The top 10 most important features based on the Extra Trees classifier.**
(TIF)

**S4 Fig Accuracy of a K-Nearest Neighbors (KNN) model as the number of neighbors k increases from 1 to 10.** The blue line represents the mean accuracy, while the red bars indicate the standard deviation. Accuracy peaks at k=6, with a lower variability as compared to k=5.
(TIF)

**S5 Fig Nanobody production and detection.** (A) Chromatogram from IMAC Purification of Nb Ab.2. The x-axis represents the elution volume (mL), while the left and right y-axes show the absorbance at 280 nm (A280) and the concentration of the elution buffer (%), respectively. (B) Fraction 13 was subjected to analysis using SDS-PAGE (17.5%). The nanobody Nb Ab.2 (19.5 kDa) migrated within the polyacrylamide gel with the expected mass, around 20 kDa. The occurrence of an additional band on the gel, positioned above the target band, is likely indicative of an incomplete removal of the pelB signal peptide from the N-terminal sequence of the nanobody [1]. The yield obtained was 0.5 mg per liter of bacterial culture. (C) Nb detection was made by using an in house His-Tag Protein ELISA assay. Plate was coated with the purified Nb during overnight. The next day, an anti-6xHis monoclonal antibody was added, followed by an HRP conjugated secondary antibody. We used the signal generated solely by the addition of the secondary antibody to the plate as our blank control.
(TIF)

**S6 Fig Production and morphological assessment of VLPs.** (A) Transmission electron microscopy (TEM) images of SARS-CoV-2 VLPs produced in Vero E6 cells 48 h after co-transfection with the plasmids encoding the virus structural proteins (M, S, E, and N) according to the 8:6:8:3 molar ratio. (B) Western-blot analysis of purified VLPs. Detection was performed using a pool of SARS-CoV-2-infected individuals.
(TIF)

**S1 Table. Interface parameter for the filtered nanobodies from the total pool of designs.**
(XLSX)

**S2 Table. Summary of Molecular Mechanics Generalized-Born Surface Area (MM-GBSA) Calculations.** The estimated free energies were decomposed to evaluate the total binding free energy components for the interaction between the two most likely docking structures of the designed Nb Ab 2 and SARS-CoV-2 RBD based on cluster size (model 0) and cluster energy (model 7).
(XLSX)

## Acknowledgments

MVFF and RCW gratefully acknowledge the Klaus Tschira Foundation for support and computational resources. MVFF acknowledges CAPES (Coordination for the Improvement of Higher Education Personnel) and the DAAD (German Academic Exchange Service). The authors thank FACEPE (Foundation for Science and Technology Development of the State of Pernambuco), CNPq (National Council for Scientific and Technological Development), FIOCRUZ (Oswaldo Cruz Foundation), and the Fiocruz Genomic Network. Partial computer

allocation was provided by the Brazilian Supercomputing Centre at LNCC. The authors thank the Protein Characterization Facility (RPT02I), Fiocruz, Brazil.

## Author contributions

**Conceptualization:** Matheus Ferraz, Roberto D. Lins.

**Data curation:** Matheus Ferraz, W. Camilla S. Adan, Tayná E. Lima, Adriele J.C. Santos.

**Formal analysis:** Matheus Ferraz, W. Camilla S. Adan, Tayná E. Lima, Adriele J.C. Santos.

**Funding acquisition:** Rafael Dhalia, Gabriel L. Wallau, Rebecca C. Wade, Isabelle F.T. Viana, Roberto D. Lins.

**Investigation:** Matheus Ferraz, W. Camilla S. Adan.

**Methodology:** Matheus Ferraz, W. Camilla S. Adan.

**Project administration:** Isabelle F.T. Viana, Roberto D. Lins.

**Resources:** Sérgio O. de Paula, Rafael Dhalia, Gabriel L. Wallau, Rebecca C. Wade, Roberto D. Lins.

**Software:** Matheus Ferraz.

**Supervision:** Sérgio O. de Paula, Rafael Dhalia, Gabriel L. Wallau, Rebecca C. Wade, Isabelle F.T. Viana, Roberto D. Lins.

**Validation:** W. Camilla S. Adan.

**Visualization:** Matheus Ferraz, W. Camilla S. Adan, Roberto D. Lins.

**Writing – original draft:** Matheus Ferraz, W. Camilla S. Adan.

**Writing – review & editing:** Matheus Ferraz, W. Camilla S. Adan, Sérgio O. de Paula, Rafael Dhalia, Gabriel L. Wallau, Rebecca C. Wade, Isabelle F.T. Viana, Roberto D. Lins.

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
