## [Decision Letter · Decision Letter 0]

11 Nov 2024

PCOMPBIOL-D-24-01697Design of nanobody targeting SARS-CoV-2 spike glycoprotein using CDR-grafting assisted by molecular simulation and machine learningPLOS Computational Biology Dear Dr. Ferraz, Thank you for submitting your manuscript to PLOS Computational Biology. After careful consideration, we feel that it is significant but does not fully meet reviewer's expectations as it currently stands. Therefore, we invite you to submit a revised version of the manuscript that addresses the points raised during the review process. Please submit your revised manuscript within 30 days Jan 11 2025 11:59PM. If you will need more time than this to complete your revisions, please reply to this message or contact the journal office at ploscompbiol@plos.org. Please include the following items when submitting your revised manuscript:* A rebuttal letter that responds to each point raised by the editor and reviewer(s). You should upload this letter as a separate file labeled 'Response to Reviewers'. This file does not need to include responses to formatting updates and technical items listed in the 'Journal Requirements' section below.* A marked-up copy of your manuscript that highlights changes made to the original version. You should upload this as a separate file labeled 'Revised Manuscript with Track Changes'.* An unmarked version of your revised paper without tracked changes. You should upload this as a separate file labeled 'Manuscript'. If you would like to make changes to your financial disclosure, competing interests statement, or data availability statement, please make these updates within the submission form at the time of resubmission. Guidelines for resubmitting your figure files are available below the reviewer comments at the end of this letter. We look forward to receiving your revised manuscript. Kind regards, William CannonAcademic EditorPLOS Computational Biology Arne ElofssonSection EditorPLOS Computational Biology

Feilim Mac Gabhann

Editor-in-Chief

PLOS Computational Biology

Jason Papin

Editor-in-Chief

PLOS Computational Biology

 **Journal Requirements:** **Additional Editor Comments (if provided):****Reviewers' comments:** Reviewer's Responses to Questions

**Comments to the Authors:**

Reviewer #1: This manuscript presents a grafting-based computational approach for designing nanobodies through combination of Rosetta-based modeling protocols, machine learning and molecular dynamics simulations. The approach was used to design nanobodies targeting the SARS-CoV-2 RBD, and the most promising one identified in-silico was experimentally tested to assess its stability and binding properties. The approach was found to be very effective in identifying a high-affinity binding nanobody without the need of high-throughput experimental screening (indeed only one nanobody was tested and was very active). This is an interesting manuscript and should be published after revising the following minor points:

- Apparently, one limitation of this method lies in identifying a single CDR loop able to host all hotspot residues identified from alanine scanning on the original antibody. Here they managed to incorporate five hotspots in one loop of the synthetic library. But often hotspots coming from different CDRs could be hard to accommodate in a single loop, hence requiring more than one loop to accommodate enough hotspots to achieve high-affinity binding. I would like the authors to comment on how to address this.

- 500,000 designs were generated for a single grafted loop, but this sounds like too many sequences for just one starting backbone configuration and only enabling mutations in the CDR loops (likely many sequences are redundant). Please comment on this.

- At the beginning of the “Sequence filtering” section it is unclear how the random forest model was generated, which is confusing. I would add a reference to the Methods section, where it is described.

- For the AF3 prediction of the nanobody-RBD complex it would be more informative to provide the RMSD and pLDDT for the predicted CDR loops, as the nanobody framework is expected to have a high-quality prediction. Have the authors generated these predictions only using one seed? With extra seeds, as reported in AF3, the predictions should increase in quality. I would recommend to check if the predictions improve in this way.

- MD simulations show that CDR3 of the designed nanobody has higher flexibility than the original loop of the parent nanobody (Fig. 4B). Is this related to differences in loop length? Also, only 1 replica has been done, which makes it difficult a fair comparison.

- The same MD simulations should be extended or done in multiple replicas to really assess that the interface remains stable. Fig. 4D shows that the RMSD increases when reaching 100 ns, but 100 ns is a short time scale to really assess this (and in only 1 replica).

- Line 297-> F333 or F377?

- Line 298-> E57 or E56?

- Line 431 looks interrupted.

- Revise the phrasing of lines 445-447.

Reviewer #2: The authors describe a pipeline to design nanobodies using a grafting strategy based on the Rosetta software suit. The strategy combines metadynamics simulations of the target in complex with a known antibody followed by Alanine scanning to determine the hotspot residues, which are then grafted onto a library of nanobodies (built by generating a pool of nanobodies by grafting distinct CDRs into a stable scaffold). Rosetta protocols are then used to introduce substitutions in the CDRs (except for the hotspots) and a costume-made ML-based function based on Rosetta metrics is used to filter the best candidates, which is followed by MD simulations. As a proof of concept, this strategy was used to target the SARS-CoV-2 spike protein receptor binding domain, yielding promising results. A selected candidate was evaluated experimentally, showing good binding affinity to the isolated RBD, which decreased when the evaluation was done in the presence of the whole spike protein. The work focuses on an important computational problem, namely the design of binders which contain loops at the interface. and presents a novel and potentially useful strategy. However, some points need to be addressed:

1- As a general comment the manuscript would highly benefit from including the neutralization assays and structural data to further show if and how the designed nanobody is able to neutralize the target.

2- It is not clear if this strategy could be applied when no structures of antibodies bound to the epitope of interest are available. This limits its applicability scope and, in particular, will tend to focus on immuno-dominant epitopes, which is one of the biases that the authors are trying to avoid. This limitation should be discussed in the manuscript.

3- Nanobodies often contain disulfide bonds to stabilize them. Was this taken into account in this strategy?

4- Regarding metadynamics simulations, there are several questions that need to be clarified:

2.1- What was the rationale behind the choice of the CVs and why was the alpha-helical and b-sheet content of the CDRs chosen as a CV, since these regions are primarily composed by loops?

2.2 – The relevant parameters used in the MetaD simulations, such as the height of the Gaussians and the frequency of deposition should be described.

2.3 – Was convergence reached in the MetaD simulations?

5- Regarding the PCA analysis performed on the MetaD and unbiased MD simulations, a detailed description of this analysis should be provided, since it is unclear which atoms were included and how the fitting was performed. Additionally it is unclear how the analysis was performed for the MetaD simulations, since a re-weighting would be necessary in this case to account for the correct Boltzmann distribution.

6- In figure 2.0, how was the average and error calculated? Following in the previous question, was any re-weighting performed given that the frames come from MetaD simulations?

7- Regarding the nanobody library, what is the size and diversity of this library?

8- Concerning the ML-based function that was used to select the best candidates, how was the model selection and hyperparameter optimization performed? Were different ML-models tested?

9- Regarding the re-design of the interface residues, why did the authors choose to use the Rosetta package for this instead of methods like Protein MPNN which seem to generate very good results in this type of tasks?

10- It could be interesting to check if using the same CDRs without grafting the hotspots from the known antibody could also generate a good binder. This could serve as a control and as an assessment of what is the role the grafted hotsposts in binding.

11- There is a problem with the text in page 13, line 431: there seems to be a mix of text from two different parts and this part of the text is incomprehensible.

Reviewer #3: uploaded as an attachment.

**Have the authors made all data and (if applicable) computational code underlying the findings in their manuscript fully available?**

Reviewer #1: Yes

Reviewer #2: **No: **The code for the ML-based filtering function should be provided

Reviewer #3: Yes

PLOS authors have the option to publish the peer review history of their article (what does this mean?). If published, this will include your full peer review and any attached files.

Reviewer #1: No

Reviewer #2: No

Reviewer #3: No

---

## [Decision Letter · Decision Letter 1]

26 Feb 2025

Dear Dr. Ferraz,

We are pleased to inform you that your manuscript 'Design of nanobody targeting SARS-CoV-2 spike glycoprotein using CDR-grafting assisted by molecular simulation and machine learning' has been provisionally accepted for publication in PLOS Computational Biology.

Best regards,

William Cannon

Academic Editor

PLOS Computational Biology

Arne Elofsson

Section Editor

PLOS Computational Biology

Reviewer's Responses to Questions

**Comments to the Authors:**

Reviewer #2: The authors have addressed all the reviewers comments.

**Have the authors made all data and (if applicable) computational code underlying the findings in their manuscript fully available?**

Reviewer #2: Yes

PLOS authors have the option to publish the peer review history of their article (what does this mean?). If published, this will include your full peer review and any attached files.

Reviewer #2: No

---

## [Editor Report · Acceptance letter]

PCOMPBIOL-D-24-01697R1

Design of nanobody targeting SARS-CoV-2 spike glycoprotein using CDR-grafting assisted by molecular simulation and machine learning

Dear Dr Ferraz,

I am pleased to inform you that your manuscript has been formally accepted for publication in PLOS Computational Biology. Your manuscript is now with our production department and you will be notified of the publication date in due course.

With kind regards,

Zsofia Freund
